# A Blockchain-Based Authentication Protocol for Cooperative Vehicular Ad Hoc Network

**DOI:** 10.3390/s21041273

**Published:** 2021-02-11

**Authors:** A. F. M. Suaib Akhter, Mohiuddin Ahmed, A. F. M. Shahen Shah, Adnan Anwar, A. S. M. Kayes, Ahmet Zengin

**Affiliations:** 1Department of Computer Engineering, Sakarya University, Serdivan 54050, Sakarya, Turkey; suaib.akhter@ogr.sakarya.edu.tr (A.F.M.S.A.); azengin@sakarya.edu.tr (A.Z.); 2School of Science, Edith Cowan University, Perth, WA 6027, Australia; mohiuddin.ahmed@ecu.edu.au; 3Department of Electrical and Electronics Engineering, Istanbul Gelisim University, Avcilar 34315, Istanbul, Turkey; afmsshah@gelisim.edu.tr; 4Centre for Cyber Security Research and Innovation (CSRI), School of IT, Deakin University, Waurn Ponds, VIC 3216, Australia; adnan.anwar@deakin.edu.au; 5Department of Computer Science and Information Technology, School of Engineering and Mathematical Sciences (SEMS), La Trobe University, Bundoora, VIC 3086, Australia

**Keywords:** Internet of Vehicles, Vehicular Ad hoc Network, blockchain, distributes storage, intelligent vehicles, Vehicular Social Networking, emergency vehicles, COVID-19

## Abstract

The efficiency of cooperative communication protocols to increase the reliability and range of transmission for Vehicular Ad hoc Network (VANET) is proven, but identity verification and communication security are required to be ensured. Though it is difficult to maintain strong network connections between vehicles because of there high mobility, with the help of cooperative communication, it is possible to increase the communication efficiency, minimise delay, packet loss, and Packet Dropping Rate (PDR). However, cooperating with unknown or unauthorized vehicles could result in information theft, privacy leakage, vulnerable to different security attacks, etc. In this paper, a blockchain based secure and privacy preserving authentication protocol is proposed for the Internet of Vehicles (IoV). Blockchain is utilized to store and manage the authentication information in a distributed and decentralized environment and developed on the Ethereum platform that uses a digital signature algorithm to ensure confidentiality, non-repudiation, integrity, and preserving the privacy of the IoVs. For optimized communication, transmitted services are categorized into emergency and optional services. Similarly, to optimize the performance of the authentication process, IoVs are categorized as emergency and general IoVs. The proposed cooperative protocol is validated by numerical analyses which show that the protocol successfully increases the system throughput and decreases PDR and delay. On the other hand, the authentication protocol requires minimum storage as well as generates low computational overhead that is suitable for the IoVs with limited computer resources.

## 1. Introduction

Internet of Vehicles (IoV) is a revolutionary addition in the field of Intelligent Transportation Systems (ITS). Typical intelligent vehicles are equipped with On Board Unit (OBU), sensors, GPS, etc. where the IoVs have communication capabilities through high-speed internet (5G/6G). Initialization of internet facility with the vehicles could be utilized to increase communication efficiency as well to increase security requirements. Vehicular Ad hoc Network (VANET) could be formed by the nearby IoVs to share information with the neighbours. IoVs could pass emergency messages (EM) which include lane change information, collision warning, congested road information, accident prevention warnings, traffic signal violation, barriers, obstacles, safe distance warning, etc. and also general messages (GM) which include different types of web services, gaming services, information of nearby gas stations, parking, restaurants, hotels, advertisements, etc. The IEEE 802.11p standard provides the Control Channel (CCH) and Service Channels (SCHs) to enable the Vehicular Social Networking (VSN) between the nearby vehicles.

Because of high mobility, it is difficult to maintain a stable connection for the IoVs during communication which results in packet drop, link blockage, and delay. Thus, to improve the communication quality at present, most of the ITS communication protocols are available to increase the efficiency of communication while the authentication, reputation, privacy, and security are getting less importance. However, in today’s world, security is an essential part of communications and establishes communication with un-authenticated IoVs are nothing but opening the path to accepting all types of security attacks. Thus, in this paper, a blockchain based authentication protocol is proposed which provides a digital signature facility to ensure confidentiality, integrity, and attack prevention supports so that IoVs can verify the authenticity of the neighbour IoVs before initiating a communication with them. Blockchain provides security services like encryption, signature, hashing, etc. Special features like decentralization, distribution, flexibility, robustness, temper-resistance, immutability, transparency, fairness, etc. help blockchain to become a prevalent tool to store various types of information for different types of applications [1,2,3,4]. By default, Ethereum blockchain uses a Elliptic Curve Digital Signature Algorithm (ECDSA), but, in the proposed method, the RSA-1024 algorithm is used because it requires a comparatively smaller time.

Managing the communication by increasing the transmission rate and decreasing the link breakage, delay and packet dropping rate (PDR) are primary challenges for VANET researchers. Several protocols are proposed by the researchers for many years to achieve better solutions. Some of the protocols are there where the IoVs get services from Road Side Units (RSUs) which require expansive infrastructural expanses [5]. IoVs could get similar services (provided by RSUs) by using the internet. On the other hand, by utilizing the bandwidth provided by IEEE802.11p, it is possible to create VSN with the neighbour IoVs and the communication areas could be increased with the help of cooperative neighbouring nodes. In this paper, a cooperative protocol is proposed to increase the communication quality. The concept of the cooperative or helping nodes is while the service provider is far from a potential receiver i.e., does not have enough signal strength to receive services from the server/sender could relay that service/information on behalf of the server. Although some overhead is created during cooperation but still the throughput provided by the cooperative node is better than typical protocols’ throughput. The proposed protocol take special care of the EMs so that it could be delivered to the receivers before 100 ms to maintain the Standard Delay Requirements (SDR) for EMs [6].

In the current novel coronavirus (COVID-19) pandemic, the busyness of the ambulance, medicine suppliers, and other related emergency IoVs become very high and thus it requires special support while providing emergency supports. Thus, in this paper, the IoVs are classified into two categories where all the emergency service providers are categorized as Emergency IoVs (EV) while the other IoVs are considered as General IoVs. This will make the authentication process of EVs faster; in addition, by utilizing VSN, it is possible to alert the neighbouring nodes so that they can provide a free passage to the EVs. All the IoVs are required to register in the Local Authentication Centers (LAC) to get the public-private key pairs which will become their identity for future communication with the blockchain. It will also help to preserve the original identity of the IoVs. All the LACs from a state are connected together as members of the blockchain and all the IoVs’ registration information are stored as transactions. By this way, all the LACs have the information of all the registered IoVs in a state. The registration process is illustrated in Figure 1.

The contribution of the paper are as follows:A blockchain based authentication schema is proposed so that, before accepting any information or service from any other source, IoVs will check the authenticity of the sender by sending a request to the blockchain. Blockchain is responsible for storing authentication information of the IoVs in a distributed fashion and supports digital signature based cryptography to ensure additional security services. IoVs have to register to their LACs to get key pairs. The public key of an IoV will be their identity during communication to preserve the privacy, and a private key will be used to send a request to the blockchain. The blockchain server will provide the reply in the form of 1 and 0, which means authentic and not-authentic, respectively.To increase the range as well as the quality of communication, a cooperative communication protocol is proposed where IoVs can become helper nodes to relay a message or service on behalf of the original sender to those IoVs who do not have a strong communication link with the sender. All the receiver IoVs will check the authenticity of the service providers as well as the helper node before accepting any message or service. An optimization algorithm is also proposed to select the best helper node.To increase the authentication speed, IoVs are divided into two types where emergency service providers are considered as EVs. Moreover, transmitted messages or services are also divided into two types and important information is considered as EMs and get priorities during transmissions and are delivered before 100 ms. To remove congested traffic for the EVs, EMs are broadcasted so that the nearby IoVs can give free passage to the EVs.

Previous research works related to authentication protocols for VANETs and cooperative VANET methods are presented with the motivation of the proposed method in Section 2. In Section 3, the structure of the blockchain based authentication schema is presented with the cooperative model of VANET. Section 4 provides the implementation details and, in Section 5, performance and security analysis of the proposed protocols are explained. Section 6 discusses the pros and cons of the proposed protocols and, finally, in Section 7, the paper is concluded by mentioning some of the possible future works.

## 2. Previous Works

The exclusive set of features available in the blockchain makes the researchers interested in utilizing it in various fields. For example, blockchain is utilized by industry 4.0 [7,8], Internet of Things (IoT) [9], Smart grid [4], transportation services like smart airports [10], smart medical [11], etc. to increase security, decentralization, trust, etc. Similarly, by collaborating with EDGE computing, cloud storage, and other mobile services, the efficiency, availability, and reliability of blockchain based systems are increased. Utilization of blockchain in ITS is also increasing to get similar advantages to provide source authentication [12], trust and reputation management [13], event and message exchange management [14], intelligent payment [15], traffic investigation [16], etc.

To ensure the authenticity of IoVs or IoVs, several authentication methods were proposed previously. Some of the researchers proposed Certificate Authority (CA) to authenticate IoVs where some of them utilize blockchain for authentication. For example, to increase the efficiency of the authentication and handover process, Lai et al. proposed SEBGMM, where blockchain is used to share information between vehicles, routers, and control databases [17]. In [18], Ali et al. use a couple of blockchains to store authorized and unauthorized vehicles information separately. However, to remove the overhead of certificate based system, Ali et al. proposed a certificate-less authentication protocol. Similarly, in [19,20], two different blockchains store certified and revoked vehicles’ information where another blockchain is there to store the transmitted messages between vehicles. Not only the vehicles but also the infrastructure’s trust information are stored in a blockchain to develop an intelligent payment system in [15]. Before any transaction, both the cash counter and the driver check the authenticity of each other, and the transaction information is also stored in the blockchain.

In [21], Javaid et al. use CA for authentication and blockchain to store the authentication information, but did not mention the sign and verification overhead. In [22], Zhang et al. proposed a blockchain based storage system where authentication information is stored with their position, location, and direction information. Additionally, to store the reputation information, all the rule violations are also added in the blockchain. However, the certificate generation, sign, and verification create high overhead.

Storing authentication information and preserving the privacy of the vehicles’ blockchain are used in [23]. To increase the security services, the method generates high computational overhead. Similarly, several other protocols like [13,24,25,26,27,28] require a lot of time to complete their authentication process. In addition, storing the authentication information of the vehicles with a proposed method by Li et al. and Salem et al. requires a lot of storage space [29,30]. All of the above mentioned protocols depending on the RSU demands expensive infrastructural supports [5]. However, typical certificate based protocols create higher overhead where the blockchain based system with light-weight encryption generates less overhead and additionally provides extra facilities of typical blockchain [12].

To increase the reliability of communication and enhance the VSN area, the efficiency of cooperation is already proven [31]. Several types of cooperation protocols are presented by the researchers to improve the performance of VANETs. In [32], a cooperative method is proposed for cluster based VANET, which is suitable only for Emergency Message Transmission (EMT). However, cooperation can be formed only if the channel is free. Woo et al. proposed a cooperative protocol applicable for EMT only, and the effect of mobility is not considered [33]. The proposed method by Taghizadeh et al. is also for EM only but unable to fulfil the SDR of 100 ms [16]. Similarly, the concurrent transmission based MAC protocol by Zhang et al. also does not fulfil the SDR and is suitable only for GMTs [34].

In [35], Zhou et al. presented a cooperative schema by using Request to Send/ Clear to Send (RTS/CTS) mechanism, but it creates additional overhead in the channel, and the possibility of collision is increased. However, the RTS/CTS based method is not suitable for EMT.

In [36], a cooperative downloading protocol was presented and, in [37], a relay broadcasting is presented to increase the availability of resources but none of them discloses the delay of their transmission protocol.

The proposed cooperation method by Bharati et al. is based on Time Division Multiple Access (TDMA) which supports point-to-point (P2P) communication only [38]. Therefore, it is not possible to broadcast EMs and communication will be stopped while there is no available time slot. However, an enhancement of the proposed method in [38] is presented in [39], where the time slots are utilized more efficiently. Similarly, Zhang et al. presented cooperation where TDMA is used with central supervision [40] and, in [41], Omar et al. presented a method called VeMAC that is also based on TDMA. However, because of mobility, VANETs are in a dynamic nature and thus TDMA protocol are not able to manage radio resources efficiently, which results in additional delay and minimized throughput [31].

Thus, a protocol that can manage both emergency and general messages, and will be efficient in terms of throughput, and have minimum delay and PDR is required. Proper resource utilization and fulfilling the SDR is also required to be considered while managing VANETs. On the other hand, to utilize cooperation efficiently, proposed management is required and cooperation should only be used when it is necessary.

### Problem Statements and Motivations

Avoiding malicious or bad intended vehicles involved in the VANET authentication of the vehicles is required. To ensure the authentication of vehicles for VANETs, several methods were presented, but blockchain based systems could be a better option with additional features like decentralization, distribution, flexibility, robustness, temper-resistance, immutability, transparency, fairness, etc. Regular certificate based protocols are not able to provide all these features together.To ensure the security of the communication, an encryption method is crucial, but blockchains usually use strong digital signature methods for encryption, which required a good amount of computational time to perform. For example, ECDSA is used by a Ethereum blockchain which required nearly 10 ms to perform one signature and verification [42]. To minimize that a light-weight encryption algorithm like RSA-1024 could be used will provide a security strength of 80 bits and require one-third the time of ECDSA [12].Preserving the privacy of the vehicles is required because identity theft could be performed by malicious entities to perform illegal activities by using it. Hiding the original identity of the IoVs’ could use public keys that can be assigned by the registration centers during registration. The real identities should be mapped with the respective public key and stored in a secured place will preserving the privacy of the IoVs.All of the transmitted messages are not the same in terms of importance. Thus, it requires to handle EMs separately by giving high priorities. Moreover, the performance of the GMT is also important in order to maintain by minimizing the delay, collision, and PDR.In the previously proposed papers, all the vehicles get equal priority, and there is no special priority for the emergency service provider vehicles. Classification of vehicles will add extra optimization, and ensuring priorities for the emergency IoVs during authentication and driving as well as classification of message or service types also provides priorities to the emergency messages during transmissions. Handling EVs separately by giving them preference while driving can help with performing emergency tasks quickly.As cooperation can increase the reliability and range of communication, it could be used for VSN. A cooperation protocol is required to be well managed to increase the throughput by minimizing the delay and PDR. Moreover, it requires handling both the general and emergency messages separately and to ensure SDR for EMs.Many of the previously proposed protocols utilize RSU for various support like computational, storage, management, etc. However, it requires additional infrastructural cost to construct RSU and maintain. On the other hand, ITS with internet facility could remove the expansive infrastructural cost of RSU by using EDGE computing services or from servers situated anywhere in the world.

These are the motivations of this research work and all the mentioned points are addressed in the proposed method and proved their efficiency in terms of performance and security.

## 3. System Structure

Intelligent vehicles with internet connectivity i.e., IoV from the nearby area, could form a VSN between themselves by performing direct or cooperative communication (when required). For the proposed method, it has been considered that all the IoVs are equipped with an OBU with data processing and wireless communication facilities, GPS, sensors, and internet connection facilities. These are the basic requirements and, without these, vehicles can not use the facilities offered by the proposed protocol.

There are four main components of the proposed system. In the first part, the registration process is discussed. Whenever an IoV requires a road permit, it has to register with the LACs. LAC then add details of the registered IoVs as blockchain transactions so that all the member LACs can get the authentication information of the IoVs. The second component is the blockchain based authentication process. The cooperative communication protocol is the third component of the system and discussed how, when, and in which situation the cooperation is required. As the fourth component, details of the VSN with the classification are explained.

### 3.1. Registration and Classification of IoVs

All of the IoVs register to their Local Authentication Centers to participate in the VANETs. LACs are responsible to generate public-private key pairs for them and to preserve their privacy, IoVs will use their public keys as the identity for all types of communications. LACs will register IoVs with all required information and generate a blockchain transaction to store those in the database. All of the LACs of a state or country will get the information immediately as a member of the blockchain. All the LACs preserve a copy of all registered IoVs’ information to form a distributed and decentralized system. The registration process is illustrated in Figure 1.

In VANET, typically all the IoVs are treated equally. However, in real-world emergency service, providers need priorities to ensure quick and efficient services. For example, in these pandemic situations, ambulances together with emergency medicine, face masks, sanitation products, COVID-19 test kits and equipment suppliers require priority services while moving. This is why the classification of IoVs is presented to give priority to the EVs. During registration or later by submitting proper documents, an IoV can be recognized as EV. To implement this, a field called type is added in the database. Other emergency service providers like fire services, civil defence, police, and VIPs could also be considered as EVs.

While driving, the EVs continuously broadcast EMs by informing the nearby IoVs that there is an existence of an EV nearby and therefore please clear the left lane and give free passage to the EV. After receiving the EM, the neighbour IoVs will clear the lane for the EVs and perform their emergency services.

### 3.2. Authentication Process

In the proposed method, there are two ways (direct and cooperative) to participate in a VSN. While driving, an IoV may receive a message or service advertisement from another source (vehicle, infrastructure, etc.). Before establishing a communication with the sender to check the authenticity of the sender, the receiver will request from the nearby LAC by sending the sender’s public key (SPK) and type (T). The server will perform a search operation in the blockchain to check the existence of the SPK in the database and send a response with the requested SPK and 0 or 1 to inform the authenticity. The receiver will take action after getting the confirmation from the server. Similarly, while getting cooperation requests or any other requests, the IoVs must check the reliability of the sender. As a decentralized system, an IoV can get the authentication checking service from all over the country and the LACs can provide instant replies within some milliseconds by just performing a lookup operation for their local storage. Optimizing the authentication process sender will check the authenticity of only the optimal helper. Details of the authentication will be explained in Section 3.4 with the cooperation details.

By default, in the Ethereum blockchain, all the communications between the blockchain server and the members are encrypted by using ECDSA [42]. However, in the proposed system, it has been replaced by a lightweight digital signature algorithm RSA-1024 which provides 80-bit security and is pretty good for light-weight devices [43].

### 3.3. Cooperation Details

By utilizing the IEEE 802.11p, IoVs can create or participate in a VSN. However because of the high velocity of the IoVs, it is always a challenge to maintain a stable network connection while communicating. By using cooperation, it is possible to increase link reliability and the efficiency of the communication [44]. However, cooperation naturally creates extra overhead, duplication, etc., thus it requires proper management to get the best from the cooperation [31]. In this paper, a mixed protocol is presented by combining direct and cooperative communication together. The dynamic nature of VANET supports both of the protocols. For random access, according to IEEE 802.11p, the CSMA/CA approach is utilized to avoid packet collisions.

To support cooperative communication protocol, some new control packets are introduced, those are NACK, KTH, SHM, WSA, WTI, and CWSA. The detail packet structure is illustrated in Figure 2 and will discuss details of it in Section 3.4.

#### Direct or Cooperative Communication?

When direct communication (DC) is possible, cooperative communication (CoC) is a waste of resource, time, etc. However, when DC is not possible because of distance or weak network connections, a helper node who has good link connections between the sender (S) and the receiver (R) could make the communication smooth and reliable. Thus, deciding to use DC or CoC is the first challenge. EMs are important and, if an IoV senses that there is an EM broadcasted in the network and somehow it could not receive it, it will initiate cooperation by broadcasting NACK. Neighbour node(s) who has a better communication link between the S and R can become a helper node to relay the EM to the receiver. In the case of GMs, if neighbouring nodes have better signal strength than a service provider, they may want to become a helper (H). The server (S) will check whether the helpers have better channel conditions than S, and it checks the helper signal for noise interference and the noise ratio (SINR) to select the optimal helper. In this way, only if the cooperation is necessary or if cooperation can provide optimized transmission will the VANET go for cooperation; otherwise, direct transmission will continue. Moreover, upon receiving any request from a neighbouring node, the IoVs first check the authenticity of that requesting IoV before establishing any kind of communication.

### 3.4. Vehicular Social Networking

IoVs can use their built-in wireless communication facility to form temporary social networking called VSN. VSN could be utilised not only for entertainment or general communication purposes but also for sharing important information or emergency messages. In the traditional IEEE802.11p MAC protocol, all of the messages get similar importance and are thus treated equally. Thus, to provide priority to EMs as well to ensure the reliability of the communication, some changes are introduced in the packet structures. To improve the communication efficiency of other general purpose messages or services, some modifications are also made in our proposed protocol. Additionally, as there is no security and privacy preserving authentication method available, a blockchain based authentication protocol is introduced so that the IoVs can get a secure environment while communicating with unknown IoVs.

#### 3.4.1. Emergency Message Transmission (EMT)

Lane change information, collision warning, congested road information, accident prevention warnings, traffic signal violation, barriers, obstacles, safe distance warning, etc. are considered as EMs. All of the nearby IoVs must know about this information to avoid fatal situations. However, because of their high speed, it is sometimes difficult to receive the EMs. A helper node may come forward to solve this issue by retransmitting the message. In IEEE 802.11p, all the transmitted messages are treated as equal and there is no special treatment for EMT. Thus, in this proposed method, NACK is introduced so that, if any IoV does not receive an EM, it can broadcast NACK to all the nearby IoVs. If no NACK is received, the transmission is considered as successful; otherwise, the sender will resend the EM to the NACK sender with the help of a helper node. The complete process is illustrated in Figure 3 by using a sample scenario.

The complete process can be described as follows:When an emergency situation comes, IoV (S) uses CCH to broadcast an EM. All of the receivers who receive that message will send the sender’s public key (SPK) with the type of the IoV to the blockchain to get the authenticity of the S. A nearby local server will handle the request and search in the database and send authorization if it is found or un-authorized.All the neighbouring nodes can sense that a message is broadcasted [38], but it may happen that, because of packet collision or weak network connections, a receiver (R) may not receive the EM. R will wait to receive it until Short Inter Frame Space (SIFS) and then broadcast NACK to its neighbours by informing that an EM transmission is unsuccessful.A NACK packet includes a unique NACK-ID, public keys of the sender (SPK) and the receiver (RPK) and the SINR between them (see Figure 2).S will wait for NACK until T${}_{s}$ (max time for successful transmission), and, if it does not receive any NACK within that time, it will consider the transmission to be successful.The IoVs who receive NACK and want to help the receiver firstly check the authenticity of the NACK sender by sending a request to the blockchain. Upon getting confirmation of the R’s authentication, it sends a Keen to Help (KTH) message to the sender by including NACK-ID, SPK and RPK. Helpers address (HPK) SINR between the helper and receiver and the packet id. KTH must be received by the sender within SIFS; otherwise, the transmission will be considered as successful, and no cooperation will be required.Even after T${}_{s}$ sender can receive KTH, which also provides information about a failed transmission. From the KTH, the sender checks the authenticity of only the optimal helper i.e., the one that has the lowest SINR from the blockchain server. Then, S sends SHM to the helper by including NACK-ID, sender, receiver and selected optimal helpers’ public keys. The sender stops receiving KTH from any other IoVs after sending the SHM.For every fail transmission, there will be different NACKs and, based on SINR between the helpers, it may be different for the same receiver. The cooperation is initiated by the receiver, which ensures that cooperation is performed only when necessary and to ensure the reliability of the communication. A blockchain based authentication service ensures that no unauthorized or fake IoVs can interface with the communication. In Figure 4, a flow chart is given to show the steps.

Blockchain based lightweight authentication protocol requires low computational time and storage. Thus, before receiving any information from any vehicle or infrastructure, the authenticity of the sender needs to be checked to avoid spamming, Sybil, unknown source, DDoS and other security attacks. However, although the authentication process is adding extra time in the transmission, it is ignorable enough and ensures that the EMs will reach all the nearby IoVs within 100 ms.

#### 3.4.2. General Message/Service Transmission (GMT)

Different types of web services, gaming services, information of nearby gas stations, parking, restaurants, hotels, advertisements, etc. are considered as GMT. It can be an IoV or RSU who offer services or want to send some information. It broadcasts the message or Wireless Access in Vehicular Environments (WAVE) service advertisement (WSA) by using the control channel. Interested IoVs may send Willing to Involve (WTI) to get the service. While an IoV is listening and planning to receive a service but facing weak network connections to communicate with the sender or server, a helper node (IoV or RSU) may come forward with better communication strength with the sender and the receiver. By this method, a cooperation process may start while transmitting GMT. The control messages will be transmitted by using CCH while the service will be transmitted by using the service channel (SCH). The complete process is illustrated in Figure 5 by using a sample scenario.

The complete process can be described as follows:Whenever a sender or server want to offer a message or service, it broadcasts WSA by using the CCH. The WSA packet consists of WSA-ID, the public keys of the sender and the receiver, ID of the Basic Service Set (BSS-ID), Service ID (SER), SINR, the Enhanced Distributed Channel Access (EDCA), SCH of the sender, etc.The interested IoVs can check the authenticity of the sender by initiating a search request to the blockchain server. After getting the positive confirmation from the authentication center, the receiver will send a WTI packet by including the WSA-ID, ID of the WTI (WTI-ID), SPK, RPK, SINR, etc.If a potential receiver is not able to send Cooperative WAVE Service Advertisement (CWSA), the server will wait for a helper who has a better connection with the receiver.A helper who wants to cooperate and have a strong connection between the sender and the receiver checks the authenticity of the receiver by using the blockchain. Then, it sends CWSA to the sender by including WTI-ID with SPK, RPK, helper’s ID (HPK), SINRm channel information, etc.A server will check the SINR of the helpers and discover the node with minimum SINR. Then, it will check the authenticity of the potential helper and send back SHM packet with the DATA. The server then transfers the data or general message or service to the helper, and the helper starts sending data to the receiver. The receiver checks the authenticity of the helper and then starts receiving by using a cooperative service.After sending SHM to a helper, the server stops receiving any other CWSA with the same WSA-ID. In Figure 6, a flow chart is given to show the steps.

## 4. Implementation

As proof of concept, the proposed blockchain based authentication protocol is implemented by using multiple virtual machines (VMs).

To implement the above-mentioned scenario in Figure 3 and Figure 5, five VMs are configured to represent sender, receiver and three potential helper IoVs. A VM is also configured to represent a nearby LAC and a blockchain server. Configurations of the VMs are presented in Table 1.

For the blockchain server machine, truffle framework [45] is used which provides a client side development environment to write, run, and test scripts for the blockchain. Additionally, it also provides network management supports. To emulate an Ethereum blockchain, a well-known emulator Ganache [46] is used. Ganache provides all facilities of blockchain with customization, logging, and debugging supports. A Node Packet Manager (NPM) [47] is used to support JavaScript, and a node server [48] is used to implement the client side.

All the machines that represent IoVs use a Metamask [49] ethereum wallet to securely communicate with the blockchain. As Metamask is platform independent and also comes as an extension to almost all types of internet explorers, IoVs using any type of machine or operating system can thus connect with the blockchain without any complexity. In the experiment, multiple platforms are used to test the system compatibility.

A server side script is written in the form of a smart contract by using the solidity programming language. The script consists of two functions: one for IoV registration and another one for searching operation. The first one is used by the LAC when a new IoV comes for registration and another one can be used by any IoVs to query for the authenticity of another IoV. For request, the requesting IoV will add the SPK with its type and send to the server. The server will reply with the requested SPK and 1, i.e., authentic if the SPK is found in the database, or 0 i.e., unknown if not found. IoVs keep a list of the requests and responses in their local storage to avoid sending requests for the same IoV’s authentication information.

## 5. Performance Analysis

Performance analysis of the proposed method is divided into two parts. In the first part, the efficiency of the cooperative transmission protocol is explained followed by the efficiency of the proposed authentication protocol.

### 5.1. Cooperative Transmission Protocol

The effectiveness of cooperative transmission to improve the reliability of communication in VANET is proved. However, it creates additional overhead and thus, without proper management protocol, it becomes inefficient. In the proposed method, some methods are applied to improve the efficiency of cooperative communication. Firstly, cooperation is used only when it is required; otherwise, the system performs direct communication and, secondly, classification of messages to ensure the priority of the EMs.

To test the VANET with randomly distributed N number of vehicles running on a multi-lane road, the normalized throughput of the proposed cooperative protocol (S) can be given as:(1)$$S=\frac{E_{p}}{T_{e}}$$
where $E_{p}$ denotes the length of the transmitted payload and $T_{e}$ denotes the slot time. From this equation, the throughput of the cooperative communication can be calculated as:(2)$$S=\frac{P_{s}P_{busy}L}{P_{h}\left[\left(1-P_{busy}\right)T_{slot}+P_{busy}P_{s}T_{s}+P_{busy}\left(1-P_{s}\right)T_{c}\right]}$$

Here,
$P_{s}$ = probability of successful transmissions,$P_{busy}$ = probability to find that the channel is busy,*L* = length of the packets,$P_{h}$ = probability of not getting a helper,$T_{slot}$ = slot time,$T_{s}$ = probability of successful transmission with cooperation,$T_{c}$ = probability of collision.

If $C_{A}$ denotes the number of cooperation attempts, PDR can be given as:(3)$$PDR={\left(1-P_{s}\right)}^{C_{A}}$$

Average packet delay can be given as:(4)$$E\left[D_{CT}\right]=T_{e-CT}\left(N-\frac{P_{fdrop}}{1-P_{fdrop}}\frac{W_{0}+1}{2}\right).$$
where $W_{0}$ and $P_{fdrop}$ denote contention window size and final packet drop probability, respectively. $T_{e}$ is the Markov state time spent for a vehicle, which can be given as:(5)$$T_{e}=\left(1-P_{busy}\right)T_{slot}+P_{busy}P_{s}T_{s}+P_{busy}\left(1-P_{s}\right)T_{c}$$

All the equations are proved and discussed in detail in [31]. Numerical analysis is presented in the next sections by comparing with the traditional MAC protocols. Data used for the analysis are presented in Table 2. For the numerical analysis, IoVs’ speed is considered as 80 km/h, and they are moving in an ideal environment. The effect of velocity is not considered in this analysis, although variations of velocity may change the performance of the system. Moreover, the effect of velocity for vehicles’ could be found in [31,50].

#### 5.1.1. Throughput

In the recommended procedure for both EM and transmission in VANETs, there is a major increase in throughput. Up to a certain extent, throughput rises, then throughput declines, as can be seen in Figure 7. For the analysis, P${}_{h}$ is considered as 0.5 in the normal case, P${}_{h}$ ≤ 0.4 as optimal and P${}_{h}$ ≥ 0.7 as the worst case. Since fewer IoVs do not cause crashes, with growing IoVs, throughput continues to increase, so, after the number of IoVs grows more, further IoVs may cause further accidents and decreases in throughput. It is also evident that the higher likelihood of having support would improve throughput, as the S-R connection would be more reliable to transmit the packet, and the transmission by cooperation will be quicker with good channel condition. Due to the availability of helpers, the throughput for the suggested protocol in optimum situations is higher than average. Nevertheless, owing to the unavailability of support, the opposite situation is viewed in the worst case. Through more aides, more collaboration benefits will be made.

#### 5.1.2. Delay

The average packet latency against the number of vehicles is seen in Figure 8. With the number of IoVs, the total packet delay grows when there are more packets to be transferred. These additional packets will compete for transmission in the same time slot for the channel, resulting in an increased channel busy likelihood as well as a probability of collision. Therefore, there is an increased average packet latency. Since this final packet drop possibility is minimized by the proposed protocol and the probability of effective transmission rises, the average packet delay is decreased.

#### 5.1.3. Packet Dropping Rate (PDR)

Figure 9 demonstrates PDR versus the number of vehicles. If the risk of crashes increases, PDR increases with the growing number of cars. The probability of packet arrival increases as there are more vehicular nodes, which would result in more accidents. The proposed protocol’s PDR benefit is important. By decreasing PDR, the proposed protocol guarantees efficient transmission. In addition, a distinction is given between the various sizes of the contention window (W${}_{0}$). When W${}_{0}$ is greater, the increased back-off period decreases the collision and reduces the failure of the packet, thus decreasing the PDR.

### 5.2. Authentication Protocol

For VANETs, safe and secure communication is a basic requirement. Generally, vehicles can create a VSN with nearby vehicles by using built-in tools according to the IEEE802.11 standards. However, the protocol is unable to provide authentication or identity preservation facility. However, typical signature based authentication protocols require comparatively high configured computers to perform. Moreover, RSU based authentication protocols require an additional infrastructural cost. Thus, an authentication protocol is proposed by utilizing blockchain technology, and, by using an internet connection, IoVs can check the authenticity of the neighbour vehicles before starting communication with them.

#### 5.2.1. Computational Overhead

By default, Ethereum uses ECDSA for signature and verification while communicating with the member nodes. However, in the proposed method, the RSA-1024 algorithm is used instead of ECDSA to minimize the execution time. It requires 1.55 ms time for signing and verifying a message (1.48 ms for signing and 0.07 ms for verifying) by a 1.5 GHz processor [51]. Therefore, the total time required to send an authentication request and get the response is 3.10 ms. Moreover, for RSA-1024, key generation requires 97 ms [43]; thus, every second, 10 keys can be generated.

Traditional ECDSA is utilized by Lin et al. in their proposed BCPPA protocol, where it requires 3.6 ms to sign and 7.2 ms to verify, i.e., 10.8 ms to complete their authentication process [24], which is approximately three times more than the proposed protocol. Several certificate-based and other types of authentication methods have been proposed previously, which also require comparatively higher time than our proposed method. For example, proposed authentication protocol by Wang et al. (B-TSCA), Azees et al. (EAPP), Zhang et al. (DSSCB), Zhang et al. (IBV), Shao et al. (IBCPPA), and Xrongxing et al. (SPRING) required 10 ms, 12 ms, 13.5 ms, 14.7 ms, 15.9 ms, and 20.1 ms, respectively [13,22,25,26,27,28]. Comparison between these protocols are illustrated in Figure 10.

IoVs can move anywhere in the country and can send a request to nearby LACs to get the authentication information of required IoVs. A member of the blockchain close to LACs stores all the IoVs information. Thus, they can provide immediate response only by searching its local storage. Searching requires ignorable time; however, proposed classification increases the efficiency of searching. If 20% of IoVs are EVs, the searching will be 80% faster for EVs and 20% faster for GVs.

#### 5.2.2. Storage Overhead

The information of every registered IoVs is stored as a blockchain transaction. Each Ethereum transaction requires 508 bytes [52]. It requires approximately 192 bytes (128-byte key + other information) to store IoV information; and the total storage requirements are 700 bytes. Thus, it requires 700 MB of space to store a million IoVs’ information. A previously proposed method by Salem et al. and Li et al. requires 1073.8 MB and 1172.3 MB to store a million vehicles’ identity information [29,30]. Therefore, the proposed method requires a lower amount of storage, and storing one billion pieces of IoV information requires only 68 GB of space.

### 5.3. Security Analysis

The primary objective of the proposed method is to ensure the security of the IoVs while communicating with each other. The security services provided by the proposed method are discussed as below.

The proposed method ensures the authenticity of the message or service provider vehicles. Whenever an IoV broadcast any EM, before accepting that message, IoVs first check the authenticity of that vehicle. Similarly, with the proposed schema, IoVs are able to ensure the authenticity of the help seeker and the helper too.RSA-1024 provides security that provides security strength of 80-bits. Thus, it requires $2^{80}$ operations to break the key that is strong enough for low power vehicles [53].IoVs are registered with their real identity, but afterwards identified by their public keys. During any type of communication, IoVs use their public keys instead of real IDs, which preserve their privacy. The original identities are stored safely in a blockchain based secured system, and an attacker will not be able to get the real identity of the vehicles even if they got the key pairs.The communication with the blockchain is encrypted by a digital signature algorithm that ensures security, confidentiality, integrity, and non-repudiation of the transaction. Encryption also prevents the message from being modified or fabricated by attackers and also from the man in the middle (MITM) attack.LACs perform physical verification of the IoVs during registration so that no fake software can perform any kind of malicious operations in the proposed system. It makes the system safe from different types of unknown source attacks, Sybil attacks and prevents any action performed by unauthorized entities. Moreover, as all the IoVs are required to be authenticated to perform any operation in the VANET, the system is safe from deadly DDoS attacks [54] as well.As multiple servers (LACs) are available to provide services in every province, the system is fully distributed and decentralized in the aspect of storage and execution.Blockchain with smart contracts added some extraordinary features like immutable storage facility, transparent storing and transactions, flexibility in accessing and managing, tamper-resistance storage, the fairness of transactions, and robustness of the stored data.

## 6. Discussion

Initiating an authentication protocol for VANET ensures a secured environment for communication. Internet supported authentication does not require additional infrastructural cost expenses. However, as it creates extra overhead, the lightweight digital signature algorithm RSA-1024 is used to ensure dependable security measures. Although, by default, Ethereum blockchain uses ECDSA for encryption, the proposed method minimizes the signature and verification time by using RSA-1024.

To ensure availability and other facilities mentioned in the previous section, a blockchain based distributed and decentralized server is proposed (hosted by LACs). All the connected LACs are the members of the blockchain to share their registered IoVs information and help to create VSN between them. The storage requirement for the server is also very low. Although the vehicles of a country are considered in this paper, it is easy to cooperate with the LACs from neighbour countries to increase the availability of the system.

The security analysis part discloses the security services as well as the attack prevention capabilities of the proposed method. However, the additional facilities provided by blockchain are also discussed there.

To give importance to the emergency information, classification for VSN is proposed that can successfully deliver data to the vehicles within SDR of 100 ms. The VANET who use traditional MAC does not have these facilities.

IoVs are generally equipped with lower computational power and storage. Moreover, they are running by using different operating systems. By considering these issues, in the proposed system, a lightweight encryption method is used so that it can be processed by the computers with minimum configuration. Additionally, as a passive member of the system, the IoVs does not require large storage facilities, and the developed system is platform independent. Thus, the IoVs do not need additional computational power or storage for the proposed system and required less time than some of the previously proposed protocols.

Classification of IoVs increase the authentication speed as well as ensure priorities to the EVs while driving. The authentication speed of both types of IoVs increases with the percentage of EVs. In the current pandemic situation and, in the future, this classification will create a great impact in the field of ITS and IoV.

Cooperation, while required, is a proved protocol to increase the reliability of communication and additionally increase the range of communication. The efficiency and performance of this protocol with the proposed optimization are proven by using numerical analysis.

## 7. Conclusions

Ensuring the identity of IoVs is an essential requirement before establishing communication in VANET. Hence, authenticity of the server is of paramount importance due to the ever growing amount of cyber attacks [55,56] on IoVs. To protect the IoVs from cyber criminals and to ensure confidentiality, security and privacy, in this paper, a blockchain based authentication protocol is proposed for cooperative VANET where IoVs will check the authenticity of other IoVs before establishing a connection. All the vehicles require internet communication capability to register to the LACs as IoV. All the LACs are members of the authentication blockchain and are able to add new IoV information as well as check the authenticity of the requested IoVs. With the help of the blockchain, LACs are connected together to form a decentralized, distributed, secured, and robust authentication service. While developing the authentication schema, IoVs’ computational, storage capabilities are considered and thus a lightweight digital signature algorithm RSA-1024 is used instead of the typical ECDSA. The performance analysis shows that it requires a minimum amount of time (only 3.1 ms) where many of the previously proposed protocols require at least 10 ms for authentication. Moreover, the storage requirements are also minimum for the LACs while the IoVs do not require additional storage capacity to use the proposed method as all the information is stored in the blockchain.

Although the authentication speed is fast, the classification of IoVs is proposed to increase the authentication speed by eight times for EVs and two times for GVs. Additionally, introducing EVs types allows them to drive more efficiently as, whenever GVs come to know about the existence of EVs, they will clear a lane for EVs. All the vehicles related to COVID-19 related help service, hospital, ambulance, medical services, fire service, emergency help, etc. will be considered as EVs and get special facility while driving.

However, because of the high mobility and dynamic nature of VANETs, it is difficult to maintain a strong or stable communication link between two vehicles. To increase the range of communication as well to ensure the reliability of the transmission link, the efficiency of cooperation is already famous. Thus, in this paper, a cooperation protocol is also proposed to increase the transmission efficiency to form VSN.

However, while too many IoVs’ information are stored in the blockchain, performance of the system will be decreased. Thus, as a potential future work, it is possible to enhance the scalability of the system—for example, a multi-level blockchain where the central server will store all the vehicles’ authentication information and the LAC only stores the information of the vehicles registered under the LAC. LACs could collect information from the national server or a central system while required. Moreover, to ensure the security and reliability of the EMs, behaviour analysis of the IoVs could be added in the future and used for reputation management of IoVs.

## Figures and Tables

**Figure 1 sensors-21-01273-f001:**
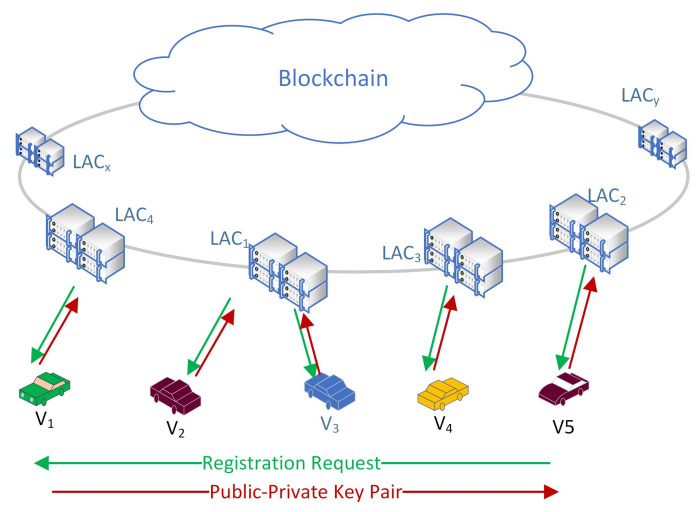
System structure and the registration process.

**Figure 2 sensors-21-01273-f002:**
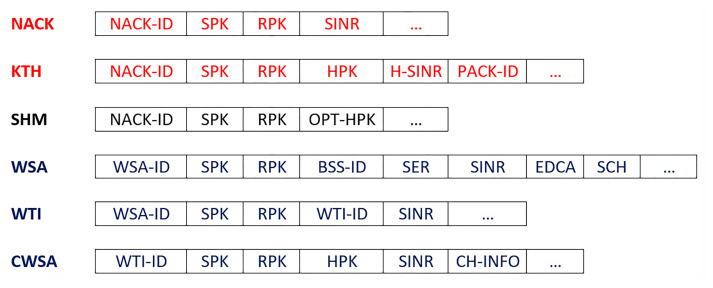
Proposed packet structure for cooperative communication.

**Figure 3 sensors-21-01273-f003:**
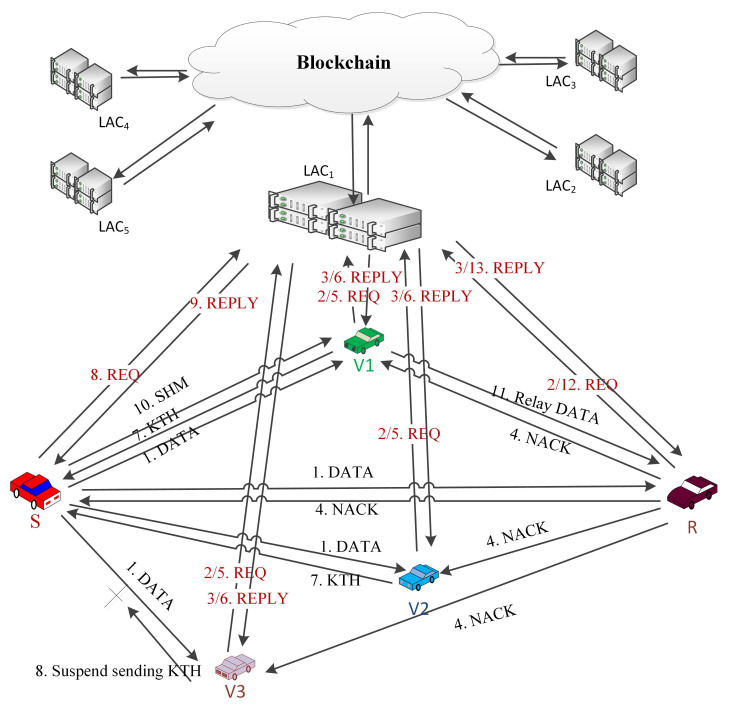
A sample scenario presenting EMT.

**Figure 4 sensors-21-01273-f004:**
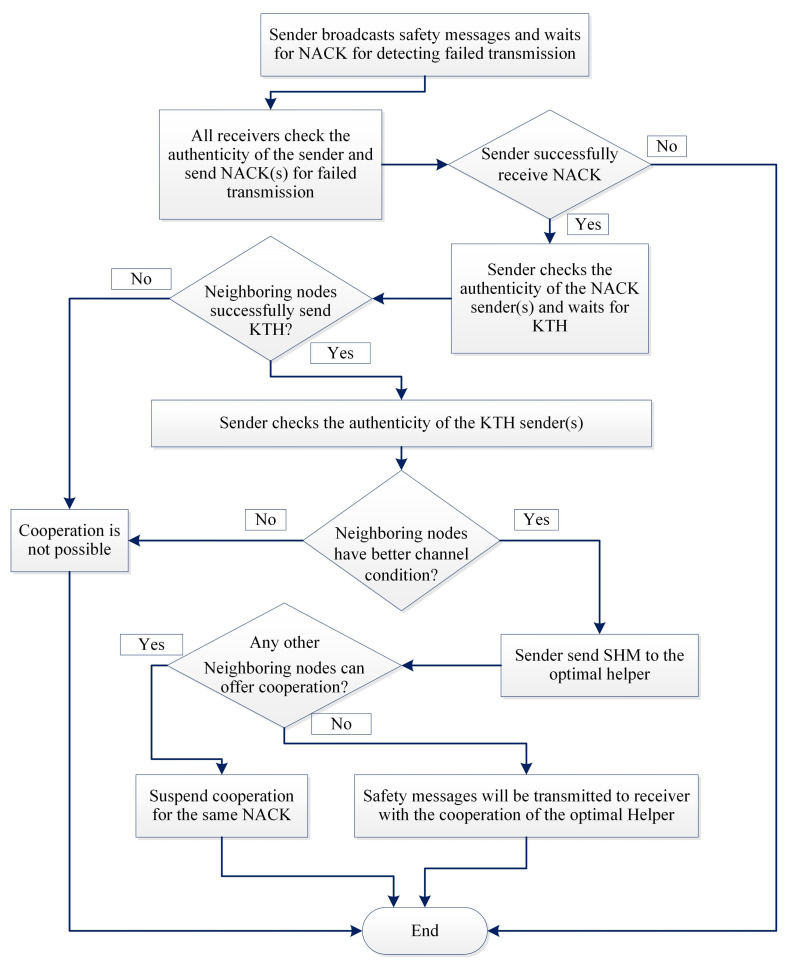
Flow chart of the proposed EMT protocol.

**Figure 5 sensors-21-01273-f005:**
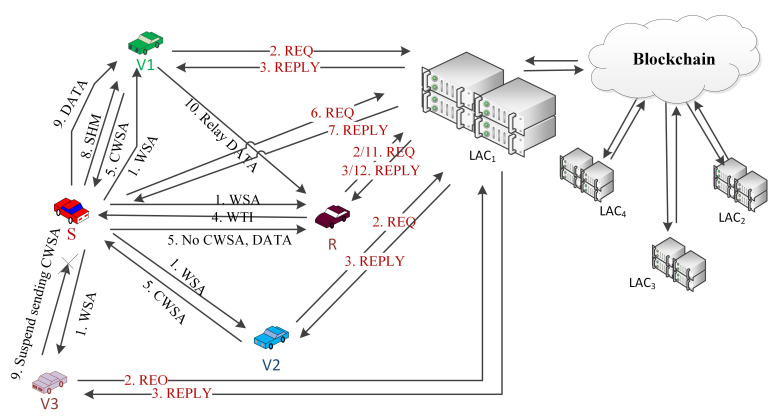
A sample scenario presenting GMT.

**Figure 6 sensors-21-01273-f006:**
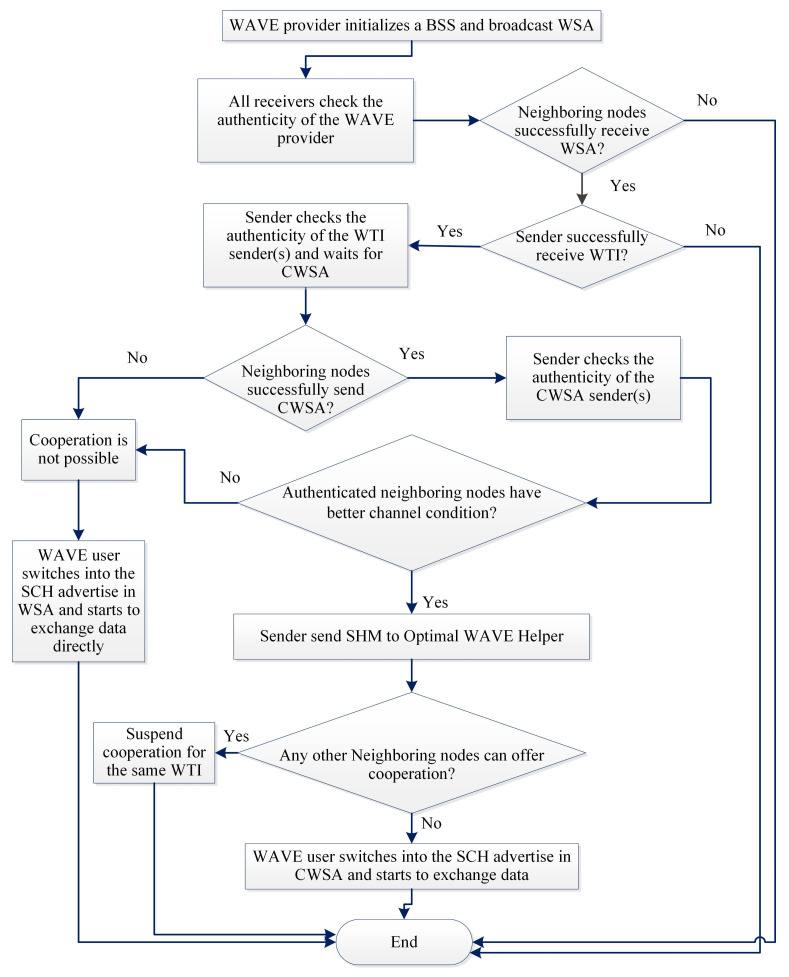
Flow chart of the proposed GMT protocol.

**Figure 7 sensors-21-01273-f007:**
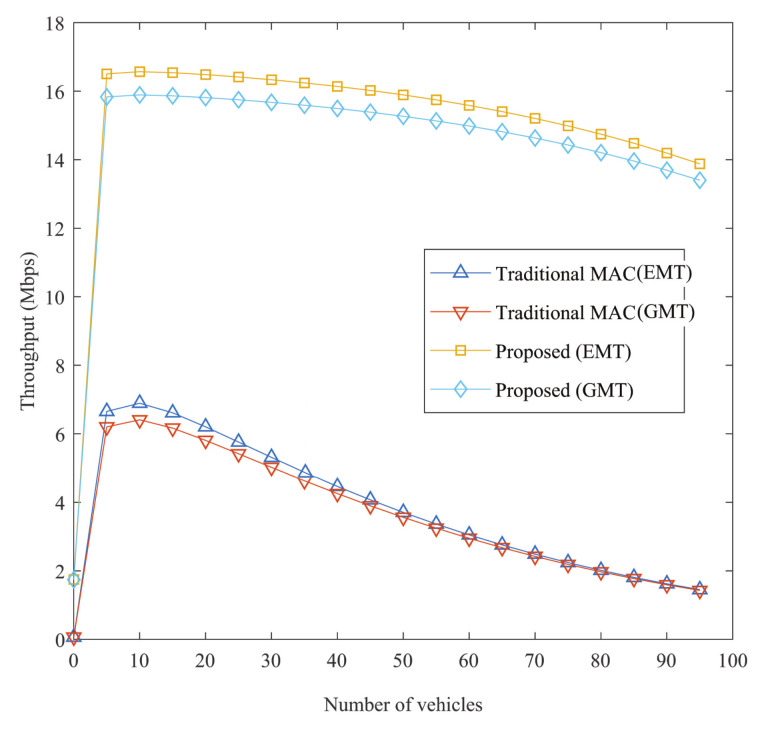
Throughput against no. of IoVs.

**Figure 8 sensors-21-01273-f008:**
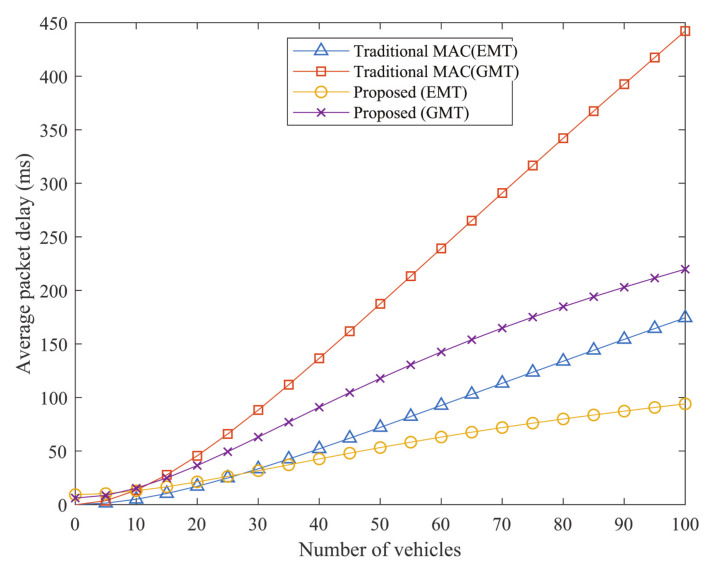
Delay versus no. of IoVs.

**Figure 9 sensors-21-01273-f009:**
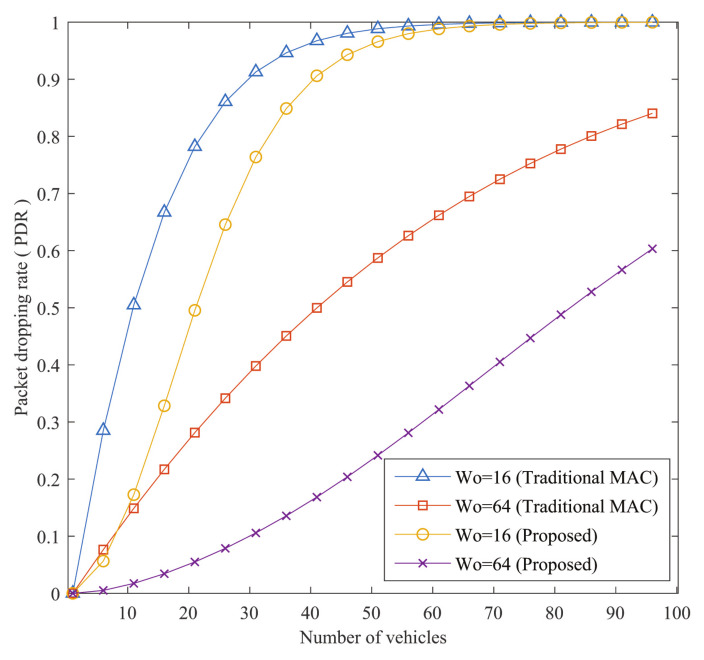
Comparison of the proposed method’s PDR with traditional MAC.

**Figure 10 sensors-21-01273-f010:**
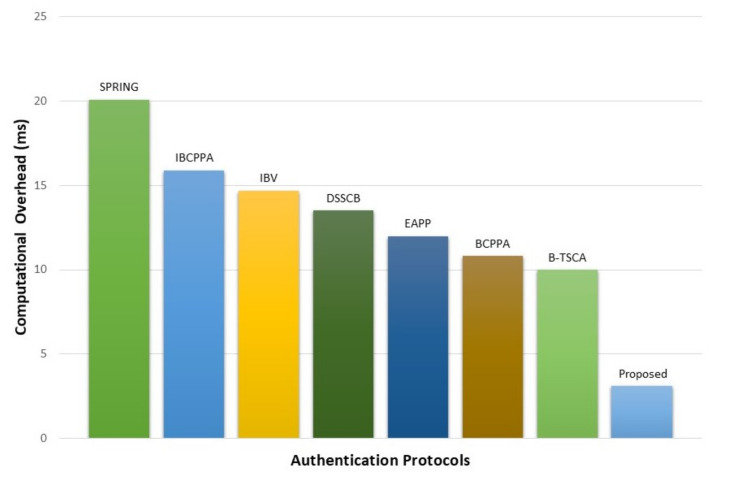
Comparison between time requirements of different authentication protocols.

**Table 1 sensors-21-01273-t001:** Implementation parameters for blockchain based authentication.

Machine	No of CPU	Memory	Storage	OS
LAC-VM	2	3 GB	30 GB	Ubuntu-18.04.4-desktop-amd64
IoV-VM${}_{S}$	1	2 GB	20 GB	Ubuntu-18.04.4-desktop-amd64
IoV-VM${}_{R}$	1	2 GB	20 GB	Ubuntu-18.04.4-desktop-amd64
IoV-VM${}_{H1}$	1	2 GB	20 GB	Ubuntu-18.04.4-desktop-amd64
IoV-VM${}_{H2}$	1	2 GB	20 GB	Windows 7 Ultimate (64 Bit)
IoV-VM${}_{H3}$	1	2 GB	20 GB	Windows 7 Ultimate (64 Bit)

**Table 2 sensors-21-01273-t002:** Sample data.

Parameter	Symbol	Value
Slot time	$T_{slot}$	20 (s)
Propagation delay	$T_{delay}$	1 (s)
DCF & Short Inter-frame space	DIFS, SIFS	50, 10 (s)
Size of the packet	$L_{h}$, L	50, 512 (bytes)
Control packets	NACK, KTH, SHM	20, 26, 24 (bytes)
Control packets	WTI, WSA, CWSA	24, 25, 27 (bytes)
Transmission range, arrival rate	R${}_{d}$, R${}_{c}$, *l*	11, 1, 0.5 (Mbps)
Contention window size	CW	64 (bytes)
Transmission range	*r*	500 (m)
Lane width	*w*	5 (m)
IoVs density	D${}_{T}$	0–0.5 (veh/m)
IoVs velocity	*v*	80 (km/h)
Average inter-vehicle distance	*b*	10 (m)

## Data Availability

Not applicable.

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
