# Peer review of "A Blockchain-Based Authentication Protocol for Cooperative Vehicular Ad Hoc Network"

_sensors, 2021, doi:10.3390/s21041273_

Round 1
Reviewer 1 Report
The authors present A blockchain based authentication schema is proposed so that, before accepting any information or service from any other source, IoVs will check the authenticity of the sender by sending a request to the blockchain. Blockchain is responsible to store authentication information of the IoVs in a distributed fashion and supports digital signature-based cryptography to ensure additional security services. IoVs have to register to their LACs to get key pairs. For this, the authors propose Blockchain strategy that is utilized to store and manage the authentication information in a distributed and decentralized environment and developed in Ethereum platform which use digital signature algorithm to ensure confidentiality, non-repudiation, integrity and preserving the privacy of the IoVs.
The paper is very interesting and well structured. The subject of communication security is a state-of-the-art subject and is not fully extended. However, in the review of the state of the art in the introduction, the reviewer did not find possible applications other than autonomous cars from the solution proposed by the authors in fields of application other than autonomous cars. In this sense, some fields of application should be included, such as Industry 4.0, with reference such as:
- Ferrer, B.R., Mohammed, W.M., Martinez Lastra, J.L., Villalonga, A., Beruvides, G., Castano, F., Haber, R.E. Towards the Adoption of Cyber-Physical Systems of Systems Paradigm in Smart Manufacturing Environments (2018) Proceedings - IEEE 16th International Conference on Industrial Informatics, INDIN 2018, art. no. 8472061, pp. 792-799. DOI: 10.1109/INDIN.2018.8472061.
Other important comment:
- Please, does not use in the sentence the first person of the plural “we”. Please, change this for passive voice and third person of the plural.
- The first figure does not appear until page 6. It is important to complement these 6 pages with a figure or table that helps to understand the text.
Author Response
Thanks to the reviewer for his review. We appreciate his effort and provided point to point reply for the review.
Best Regards,
A F M Suaib Akhter

Reviewer 2 Report
The proposal introduces a novel blockchain based secure and privacy preserving authentication protocol for Internet of Vehicles (IoV), where DLT is utilized to store and manage the authentication information in a distributed and decentralized environment enabling by Ethereum digital signature algorithm to ensure confidentiality, non-repudiation, integrity and preserving privacy. The paper addresses a raising topic, where the adoption of DLT present a promising solution for state-of-the-art AthZ/AthN challenges at ICT edge operation. However, it is also a widely addressed topic, which a large and growing scientific biography. The paper contents are adequately written and structured, fitting the journal scope, but requiring a clearer representation of the problem statement. The achieved conclusions were validated by analytical results. Overall, this reviewer suggests its acceptation after properly addressing the following minor recommendations:
- Although the Introduction section properly addresses the major contributions of the proposal, it should be remarked their differentiating aspects regarding similar state of the art solutions. Note that the application of DLT on related environments (edge computing, industry 4.0, digitalization of transport infrastructure: smartports, smartairports, etc).
- It will be interesting to include a description of the proposal design principles and problem statement: primarily/secondary goals, pre-requirements, assumed limitations, etc….as well as a formalization of the problem to be solved.
- Discussions and conclusions should provide a more critical view, highlighting the pros of the proposal (as commented), but also the aspects that require further enhancement and research. They should be explicitly indicated at the Conclusions section
Author Response

(The authors gave the same response as above.)

Reviewer 3 Report
The given article introduced the new and interesting idea of how the blockchain technology should be used in the VANET network. There is described new blockchain based authentication protocol to store and manage the authentication information in a distributed and decentralized environment.
The proposed algorithm used to optimize the performance of the authentication process IoVs. The main idea of the proposed authentication algorithm is hard readable and understandable. The main limitation of the article is performance analysis as well. The collected result could be more discussed, analysed and compared with existing solutions. I have found some English grammar and mistake in the manuscript:
- English grammar errors (pp. 6 - Figure 6 etc.). The article should be carefully improved since there are so many typos and grammatical errors.
- It is ambiguous how long keeps the blockchain server the stored data for the algorithm. I think the „ageing“ of the stored information can affect the performance of the proposed algorithm.
- What kind of propagation model has been used for simulation? Is it an ideal environment? How the speed of IoV vehicles was changed, randomly?
- There is no analysis of how the velocity (speed) of the IoVs can affect the performance of the proposed algorithms in the sense of parameters throughput, delay and PDR (Packet Dropping Rate). The speed can significantly influence performance. The performance analysis in section 5.3 Security analysis is vague. It is not clear how the results are collected and analysed. These parts need to be enhanced.
Author Response

(The authors gave the same response as above.)

Round 2
Reviewer 1 Report
The authors have modified the article according to the reviewer´s comments. For this, the paper has improved the scientific soundness and significance of content.
Reviewer 3 Report
The authors have addressed most of our former concerns and my recommendation is acceptance.